



A Novel Method for Quantifying the Contribution of Regional Transport to PM2.5 in Beijing
(2013-2020): Combining Machine Learning with Concentration-Weighted Trajectory Analysis
Kang Hu[1], Hong Liao[1], Dantong Liu[2], Jianbing Jin[1], Lei Chen[1], Siyuan Li[2], Yangzhou Wu[3],
Changhao Wu[4], Shitong Zhao[2], Xiaotong Jiang[5], Ping Tian[6,7], Kai Bi[6,7], Ye Wang[8], Delong
Zhao[6,7]
[1]Jiangsu Collaborative Innovation Center of Atmospheric Environment and Equipment
Technology, Jiangsu Key Laboratory of Atmospheric Environment Monitoring and Pollution
Control, Nanjing University of Information Science & Technology, Nanjing 210044, China.
[2]Department of Atmospheric Sciences, School of Earth Sciences, Zhejiang University,
Hangzhou 310058, China.
[3]Guangxi Key Laboratory of Environmental Pollution Control Theory and Technology, Guilin
University of Technology, Guilin 541004, China.
[4]Institute of International Rivers and Eco-security, Yunnan University, Kunming 650091, China.
[5]College of Biological and Environmental Engineering, Shandong University of Aeronautics,
Binzhou, 256600, China.
[6]Beijing Key Laboratory of Cloud, Precipitation and Atmospheric Water Resources, Beijing
100089, China.
[7]Field Experiment Base of Cloud and Precipitation Research in North China, China
Meteorological Administration, Beijing 100089, China.
[8]Key Laboratory of Meteorological Disaster, Ministry of Education (KLME)/Joint
International Research Laboratory of Climate and Environment Change (ILCEC)/
Collaborative Innovation Center on Forecast and Evaluation of Meteorological Disasters (CIC-
FEMD), Nanjing University of Information Science and Technology, Nanjing 210044, China.


Corresponding author: Hong Liao (hongliao@nuist.edu.cn)






**Abstract**
Fine particulate matter ($PM_{2.5}$) is closely linked to human health, with its sources generally
divided into local emissions and regional transport. This study combined concentration-
weighted trajectory (CWT) analysis with the HYSPLIT trajectory ensemble to obtain hourly-
resolution pollutant source results. The Extreme Gradient Boosting (XGBoost) model was then
employed to simulate local emissions and ambient $PM_{2.5}$ in Beijing from 2013 to 2020. The
results revealed that clean air masses influencing the Beijing area mainly originated from the
north and east regions, exhibiting a strong winter and weak summer pattern. Following the
implementation of the Air Pollution Prevention and Control Action Plan (Action Plan) by the
Chinese government in 2017, pollution in Beijing decreased significantly, with the most
substantial reduction in regional transport pollution events occurring in the west region during
summer. Regional transport pollution events were most frequent in spring, up to 1.8 times
higher than in winter. Pollutants mainly originated from the west and south regions, while
polluted air masses from the east showed the least reduction, and the proportion of pollution
sources from this region is gradually increasing. From 2013 to 2020, local emissions were the
main contributors of pollution events in Beijing. The Action Plan has more effectively reduced
pollution caused by regional transport, particularly during autumn and winter. This finding
underscores the importance of Beijing prioritizing local emission reduction while also
considering potential contributions from the east region to effectively mitigate pollution events.
**Keywords:** Fine particulate matter ($PM_{2.5}$); concentration-weighted trajectory (CWT);
XGBoost model; regional transport



## 1. Introduction

Ambient fine particulate matter ($PM_{2.5}$, with particle aerodynamic diameter $\leq 2.5$ μm) is influenced by both natural sources, such as volcanic eruptions, tsunamis, and forest fires, and anthropogenic emissions, including fuel combustion, transportation, and industrial production. Anthropogenic emissions dominate the long-term trend of air pollution (Zhang et al., 2019; Cheng et al., 2019). Numerous epidemiological studies have found that $PM_{2.5}$ can significantly damage human health by exacerbating respiratory and cardiovascular diseases (Bartell et al., 2013; Brauer et al., 2012; Pascal et al., 2014), and also has an impact on weather and climate change (Wang et al., 2014). China's rapid and energy-intensive development over the past several decades has led to severe air pollution and negative public health impacts (Huang et al., 2014). Consequently, controlling pollution and reducing $PM_{2.5}$ concentrations became an urgent issue in China. While meteorological variations caused about 16% of the ambient $PM_{2.5}$ decline during 2013-2017 (Zhang et al., 2019), the uncertainty in reducing $PM_{2.5}$ through meteorological conditions is substantial, and the magnitude of the decrease is not dominated by human actions. Thus, the primary means of controlling $PM_{2.5}$ relies on reducing anthropogenic emissions. To address this issue, the Chinese government implemented the Air Pollution Prevention and Control Action Plan (denoted "Action Plan") from 2013 to 2017 and the Blue Sky Protection Campaign from 2018 to 2020, which effectively controlled anthropogenic emissions and reduced ambient $PM_{2.5}$ concentrations.

The concentration of $PM_{2.5}$ can be attributed to local emissions and regional transport. Several methods, such as the HYSPLIT model (Draxler and Rolph, 2010), can be used to distinguish pollutant sources. Wu et al. (Wu et al., 2021) used the HYSPLIT model to simulate the 24-hour backward trajectory in Zhoushan, and identified continental air masses that spent more than 5% of the previous 24 hours over the continent region, while the remaining air masses were identified as oceanic-influenced air masses. Ding et al. (Ding et al., 2019) employed a backward trajectory ensemble to analyze the sources of air masses in Beijing during the study period, finding that air masses with high concentrations of black carbon (BC) mass mainly came from the south and southeast regions. Cluster analysis on backward trajectories can be used to obtain the main direction of aerosols over a period of time, allowing for the analysis and determination of dominant air mass directions. For instance, Li et al. (Li et al., 2022) divided the sources of air masses in the Wuhan area from October to November 2019 into short transport distance, northbound air masses, and regional transport from the northeast and some coastal areas.

The HYSPLIT model results are mainly used to view air mass trajectories, making it difficult to directly determine the sources of pollutants. Potential source contribution function (PSCF) and concentration-weighted trajectory (CWT) analyses based on backward trajectories can be used to identify the sources of pollutants through conditional probability results. Hu et al. (Hu et al., 2020) used weighted PSCF to analyze the sources of air masses with different levels of pollution in Beijing and found that polluted air masses from the southwest were an important source of high-level advections during the study period, while light pollution was often accompanied by the regional transport originating from the northeast region. Wu et al. (Wu et al., 2024) used CWT to analyze the sources of pollution in Zhoushan and found that pollutants in Zhoushan are influenced by both local emissions and regional transport. There are no obvious high pollution areas, while in other seasons, $PM_{2.5}$ mainly originates from southern Jiangsu and



Shanghai. However, these studies relied on standard HYSPLIT trajectory results, which have
lower temporal resolution, limiting the accuracy of pollutant source identification.
The Lagrangian air pollution dispersion model, Numerical Atmospheric-dispersion Modelling
Environment (NAME) (Jones et al., 2007) can determine the source of polluted air masses by
simulating particulate concentrations within each grid point using Monte Carlo methods,
followed by 3-D trajectories of plume basins. Liu et al. (Liu et al., 2020) used the NAME model
to study the sources of air masses in Beijing during the winter of 2019 and divided them into
local emissions and regional transport to analyze the convective mixing process of BC under
the influence of local emissions. However, due to limitations in computing resources, the
NAME model is difficult to use for obtaining long-term emission source analysis results.
Multiple methods can be used to predict $PM_{2.5}$ concentrations, such as statistical models (e.g.,
linear mixed-effect models and generalized additive models) (Fang et al., 2016; Ma et al., 2016),
chemical transport model (CTM)-based algorithms (Geng et al., 2015; Kong et al., 2021),
physical models (Lin et al., 2018), and recently emerging machine learning models, including
Extreme Gradient Boosting (XGBoost) and Random Forest (Liang et al., 2020; Wei et al., 2021;
Xiao et al., 2018; Xue et al., 2019; Huang et al., 2021). Geng et al. (Geng et al., 2021)  used
satellite observations of aerosol optical depth (AOD) and meteorological data combined with
the XGBoost model to explore the long-term variations of $PM_{2.5}$ caused by changes in
meteorological conditions from 2000 to 2018. Kleine Deters et al. (Kleine Deters et al., 2017)
demonstrated the relevance of statistical models based on machine learning for predicting $PM_{2.5}$
concentrations from meteorological data. This method of predicting aerosol concentrations
using only meteorological data has been widely used (Asadollahfardi et al., 2016; Zeng et al.,
2021). For instance, Grange et al. (Grange et al., 2018) used meteorological data, synoptic scale,
planetary boundary layer height (PBLH), and time variables to explain daily $PM_{10}$
concentrations in Switzerland. In summary, machine learning models have achieved high
accuracy in estimating and predicting $PM_{2.5}$ concentrations and have high use value, and the
rise of machine learning methods has also provided feasibility for quantifying the contribution
of regionally transported air masses.
In this study, we combined CWT analysis with the HYSPLIT trajectory ensemble to obtain
hourly-resolution $PM_{2.5}$ source results and used this approach to distinguish between local
emissions and regional transport. Predictive XGBoost models were developed for Beijing using
meteorological data and time variables to explain local and ambient $PM_{2.5}$ concentrations. By
combining these two methods, the contribution of regional transport to $PM_{2.5}$ in Beijing can be
quantified.

**2.  Materials and methods**
2.1 Site and instrumentation
The $PM_{2.5}$ data (Fig. 1a) were obtained from in situ air quality monitoring conducted by the
China National Environmental Monitoring Center from 2013 to 2020. The monitoring station
is located in Haidian Wanliu (39.96°N, 116.29°E), situated in the central urban area of Beijing.





Meteorological data, including temperature, relative humidity, pressure, precipitation, wind
speed, and PBLH, were sourced from the European Centre for Medium-Range Weather
Forecasts        (ECMWF)        ERA5        hourly        reanalysis        dataset
(https://cds.climate.copernicus.eu/datasets).

2.2 Air mass source
The air mass trajectory data were obtained from the 1°×1° horizontal and vertical wind fields
of    the    Global    Data    Assimilation    System    (GDAS)    reanalysis    products
(ftp://arlftp.arlhq.noaa.gov/pub/archives/gdas1), available every 3 hours. The HYSPLIT
trajectory ensemble was used to generate 27 equally probable 24-hour backward air mass
trajectories for the target point (39.96°N, 116.29°E, 250 m a.s.l.) in every hour by using PySplit
(Cross, 2015). Given the equal probability of air masses being transported to the target point
for each trajectory in the HYSPLIT trajectory ensemble, a conditional probability CWT
analysis was applied to determine the hourly source area of pollution.
In the CWT analysis method, each grid point is assigned a weight (equation 2), and the
contribution of each grid point to the pollutant concentration at the target site is calculated using
the air mass residence time and pollutant concentration (Hopke et al., 1993; Polissar et al., 1999;
Xu and Akhtar, 2010) (equation 1). The grid point resolution was set to 0.25°×0.25° for this
study. In equations 1, $C_{ij}$ is the average weighted concentration at grid point $(i, j)$, $l$ is the
trajectory index, $M$ represents the total number of trajectories, $C_l$ is the PM$_{2.5}$ concentration
corresponding to the target site, and $\tau_{ijl}$ is the residence time of trajectory $l$ passing through
the grid point. In calculation, the number of trajectories falling on each grid point is used instead
of the residence time.
$$C_{ij} = \frac{\sum_{l=1}^{M} C_l \times \tau_{ijl}}{\sum_{l=1}^{M} \tau_{ijl}} \times W(n_{ij}) \qquad (1)$$

$$W(n_{i,j}) = \begin{cases} 1.00, \ 3n_{ave} < n_{ij} \\ 0.70, 1.5n_{ave} < n_{ij} \leq 3n_{ave} \\ 0.40, n_{ave} < n_{ij} \leq 1.5n_{ave} \\ 0.17, n_{ij} < n_{ave} \end{cases} \qquad (2)$$

where $n_{ij}$ represents the number of trajectories that fall within the grid point, and $n_{ave}$
represents the average number of trajectories passing through each grid point.
The potential source contribution to PM$_{2.5}$ at the target site was investigated by segregating the
region where the backward air masses had passed into five parts: local (which is a region around
central Beijing, 115.3~117.5°E, 39.4~41°N); north region (the northern plateau at 108~117.5°E,
41~43°N); west region (the western plateau at 108~115.3°E, 34~41°N); south region (the
southern plain at 115.3~120°E, 34~39.4°N); and east region (the eastern plain at 117.5~120°E,
39.4~43°N). The concentration is integrated over each grid point in each segregated region
obtained from the CWT analysis, and the contributions of each air mass fraction are obtained.
The region with the highest contribution is used to determine the dominant source of air masses
in Beijing at each time, classifying the overall air mass sources into local emissions (Fig. 1g)
and regional transport (Fig. 1h).






2.3 Deriving the long-term local emission and ambient $PM_{2.5}$
An XGBoost model is employed to derive the local and ambient $PM_{2.5}$ results. The
hyperparameters used in the model include the maximum number of boosting iterations,
learning rate, maximum depth of a tree, minimum sum of instance weight needed in a child,
subsampling ratio of a training instance, and subsampling ratio of columns when constructing
each tree. The input parameters for the XGBoost model comprise meteorological variables
(temperature, relative humidity, wind speed, surface pressure, and precipitation) and temporal
parameters (year, month, day of the week, and day of the year), as referenced from Xu et al.
(Xu et al., 2023). Additionally, PBLH, which has been shown to significantly impact pollutant
concentrations in previous observational (Su et al., 2018; Miao and Liu, 2019; Miao et al., 2019)
and machine learning studies (Xiao et al., 2021; Li et al., 2017b; Shen et al., 2018), was included
as an input parameter. For the machine learning process, data from 2013 to 2019 were used for
training the XGBoost models, while data from 2020 were used for model validation.
The relatively small proportion of high-concentration $PM_{2.5}$ can lead to underestimation of
high-concentration events in the model results (Wei et al., 2020). To address this issue, a high
$PM_{2.5}$ indicator was defined as a daily average $PM_{2.5}$ concentration exceeding the monthly
average plus twice the standard deviation. In this study, original high $PM_{2.5}$ indicators accounted
for 6% of the data points during the period dominated by local and ambient $PM_{2.5}$. To balance
the proportion of high-concentration $PM_{2.5}$ in the entire database, the Synthetic Minority Over-
sampling Technique (SMOTE) (Torgo, 2011) was applied during data preprocessing. SMOTE
artificially generates new synthetic samples along the line between high-concentration data
points and their selected nearest neighbors, effectively oversampling the high-concentration
data. As a result, the proportion of high $PM_{2.5}$ indicators increased to 21% and 22% for local
and ambient $PM_{2.5}$, respectively.
Hyperparameter optimization and performance evaluation of the model were conducted using
fivefold cross-validation (CV). In this approach, 20% of the data is randomly selected for model
testing, while the remaining 80% is used for training. This process is repeated five times,
ensuring that each record is used once as testing data. The coefficient of determination ($r^2$) was
employed to assess the correlation between the XGBoost model predictions and observed
values, while the root mean square error (RMSE) was used as a performance evaluation statistic.
After obtaining the relation between the input parameters and $PM_{2.5}$, we are able to derive the
hourly local and ambient $PM_{2.5}$ once all long-term input parameters (Fig. S2).
**3 Results and discussion**
3.1 Evaluation of the XGBoost $PM_{2.5}$ prediction model
During the model validation process, the XGBoost model results for ambient $PM_{2.5}$ (Fig. 2a2)
demonstrated an $r^2$ of 0.74 and an RMSE of 20 µg m$^{-3}$ when compared to observations. The
XGBoost model results for local $PM_{2.5}$ exhibited an $r^2$ of 0.78 and an RMSE of 21 µg m$^{-3}$. An
analysis of the $PM_{2.5}$ frequency distribution in Beijing revealed a strong agreement between the



XGBoost model results and observations for both ambient and local $PM_{2.5}$. As illustrated in Fig. S1, local and ambient $PM_{2.5}$ in Beijing display a distinct seasonal variation, with higher values in winter and lower values in summer. However, the transport of clean air masses from the north diminishes the seasonal variation characteristics of ambient $PM_{2.5}$ in Beijing, making winter pollution less prominent compared to other seasons.

Fig. S2 reveals that ambient pollution events ($PM_{2.5} > 75$ µg m$^{-3}$) in Beijing are primarily influenced by air masses originating from the south and west, particularly under the control of westward air masses. With the exception of December (Fig. 3b1), westward air masses often bring higher monthly average $PM_{2.5}$ to Beijing. Air masses originating from the south region can also transport more pollutants to Beijing (Fig. S2). However, unlike the high-frequency polluted air masses from the west, southward air masses are associated with higher $PM_{2.5}$ concentrations, particularly during autumn and winter (Fig. 3c1). This phenomenon can be attributed to the higher pollution levels in Hebei and Shandong provinces compared to Beijing during these seasons, as verified by AOD observations from Moderate Resolution Imaging Spectroradiometer (MODIS) on the Aqua satellites over Eastern China (Zhang and Reid, 2010; Hu et al., 2018) (Fig. S4). Notably, in contrast to westward transport, air masses from the south region in February predominantly exhibited a cleaning effect on Beijing, even before 2017 (Fig. S2b). This can be explained by the occurrence of these transport processes during or shortly after the Spring Festival, a period characterized by extremely low anthropogenic emissions, resulting in lower ambient $PM_{2.5}$ compared to local emissions in the megacity of Beijing. Following the implementation of the Action Plan, the polluted air masses from the south region transitioned from carrying higher $PM_{2.5}$ to levels close to local emission concentrations in Beijing, leading to a more equal contribution to pollution and clean events in the area (Fig. S3c1).

### 3.2 Impact of clean air masses from transported regions on $PM_{2.5}$ in Beijing

In this study, clean air masses are defined as those associated with ambient $PM_{2.5}$ in the Beijing area that are lower than the concentrations resulting from local emissions, as illustrated below the dashed line in Fig. 3a1-d1. This study reveals that clean air masses predominantly originate from the east and north regions during the period 2013-2020, which is consistent with previous studies (Zhang et al., 2018; Hu et al., 2020). Clean air masses from different directions exhibit similar seasonal variations in their ability to reduce locally emitted pollution in Beijing, with a strong reduction effect in winter and a weaker effect in summer (Fig. 3a2-d2). This phenomenon is closely related to the seasonal variations in pollutant emissions. Due to the combined influence of increased residential emissions from heating activities and meteorological conditions in Beijing during autumn and winter, local $PM_{2.5}$ in Beijing presents higher concentrations. Consequently, the influx of clean air masses results in a more pronounced reduction in $PM_{2.5}$ during these seasons. The weaker attenuation effect of $PM_{2.5}$ transported from the south region during December and January can be attributed to the high-frequency and high-concentration pollution contributions from air masses originating in this region during this period.




Due to a significant reduction in anthropogenic emissions after 2017, the attenuation of $PM_{2.5}$
concentrations by clean air masses from all directions was significantly lower than before 2017
(Fig. S5a2-d2). Compared to the period prior to 2017, the mean attenuation of $PM_{2.5}$
concentrations in Beijing decreased by 3, 10, 3, and 7 µg m$^{-3}$ ($p < 0.01$) for air masses
originating from the north, west, south, and east regions, respectively.
3.3 Variations in Beijing $PM_{2.5}$ concentrations under transport-induced pollution events
Transport-induced pollution events in Beijing are defined as the occurrence of ambient $PM_{2.5}$
exceeding both local $PM_{2.5}$ and the light pollution standard (75 µg m$^{-3}$). Fig. 4a1-d1 demonstrate
that the monthly variation of $PM_{2.5}$ in Beijing generally follows a unimodal pattern, with higher
values in winter and lower values in summer, except when under the influence of eastern air
mass transport. This phenomenon is closely related to the seasonal variations in anthropogenic
emissions in China and the characteristics of climate change (Renhe et al., 2014; Li et al., 2017a;
Zhang et al., 2015). The overall $PM_{2.5}$ in Beijing under the influence of eastward pollution air
masses exhibits a bimodal distribution, with frequent high-concentration pollution events
occurring in January and October. Even after the effective control of anthropogenic emissions
in 2017, a second peak of high-concentration pollution persists in October (Fig. 4d2). Fig. 4a2-
d2 illustrate the effectiveness of the Action Plan in controlling pollutant concentrations in the
Beijing area. Since 2017, $PM_{2.5}$ in Beijing has been significantly lower than the values observed
before 2017 during transport-induced pollution events. Moreover, during January and from
June to September, there were periods when the regional transport of polluted air masses from
a fixed direction did not contribute to pollution events in Beijing.
An analysis of the proportion of transport-induced pollution events from different regions in
Beijing (Fig. 5) shows that after the implementation of the Action Plan in 2017, the number of
pollution events dominated by regional transport decreased significantly. From spring to winter
(defined as January-February and December of the same year in this study), the largest decrease
in transport-induced pollution events occurred in the north, west, west and south regions in each
season, with the lowest decrease occurring in the east region during winter. Among all regions,
the east region exhibited the smallest decrease in transport-induced pollution events. This is
likely due to the fact that eastward air masses have already been contributing a significant
amount of clean air to the region.
The temporal variation in the number of transport-induced pollution events from different
regions (Fig. S6) revealed that air masses transported from the west region contributed to the
most frequent pollution events in each season except summer. The highest number of events
occurred in spring 2016 (322), autumn 2016 (375), and winter 2017 (308). Summer transport-
induced pollution events were mainly influenced by polluted air masses transported from the
south, with a gradual decrease in the number of events over the years. Although pollution events
in Beijing primarily occur in autumn and winter, this study found that after 2017, the season
when Beijing was most affected by transport-induced pollution events was spring, contributing
a total of 685 pollution events, while autumn and winter contributed 266 and 392 events,
respectively. The impact of polluted air masses on summer transport was minimal, with only
215 occurrences.





Fig. 5a shows that in spring, transport-induced pollution events in Beijing were mainly
dominated by polluted air masses transported from the west and south. The highest proportion
of regional transport events from the west occurred in 2016, reaching 68%, while the highest
proportion of southward transport-induced pollution events occurred in spring 2020. The
increased frequency of pollution air masses transported from the south after 2017 can be
attributed to the effective control of anthropogenic emissions, resulting in a decrease in $PM_{2.5}$
transported from various regions, especially from westward sources (Fig. S6a). The decrease
in the proportion of pollution events transported from the west, which originally accounted for
a large proportion, led to an increase in the contribution of remaining incoming air masses to
Beijing.
Before 2017, transport-induced pollution events in Beijing during summer were mainly
affected by polluted air masses from the south. Even in 2015, when the proportion of transport-
induced pollution events from south region was lowest during the entire period, it still
accounted for 50% of the total number of transport-induced pollution events that year. However,
after the implementation of the Action Plan, the proportion of transport-induced pollution
events from the south region gradually decreased from 57% to 25%. Meanwhile, pollution air
masses originating from the east increasingly dominated the occurrence of pollution events in
Beijing.
Transport-induced pollution events in Beijing mainly originated from the west and had the
highest contribution proportion in autumn before 2019 (except for 2013, when the contribution
proportion was 34%, second only to southward air masses at 35%). After 2019, the contribution
of eastward air masses became dominant in autumn. In winter, polluted air masses from the
west were the main source of transport-induced pollution events. In 2020, the east region,
previously believed to contribute significant amounts of clean air, substantially contributed to
transport-induced pollution events across various seasons. This finding may prompt Beijing to
prioritize emission reduction in the east region when implementing future joint prevention and
control measures.

## 4 Conclusion

This study combined a machine learning method and Concentration-Weighted Trajectory
(CWT) analysis to derive local emissions and ambient observed $PM_{2.5}$ in Beijing from 2013 to
2020, thus the contribution of regional transport to $PM_{2.5}$ in Beijing can be quantified. The
impact of clean air masses (defined as those with ambient $PM_{2.5}$ concentrations lower than local
emissions) mainly originated from the east and north regions. These clean air masses from
different directions exhibited similar seasonal variations in their ability to reduce ambient
pollution in Beijing, with a stronger reduction effect in winter and a weaker reduction effect in
summer.
Except for the regional transport from the east region, the seasonal variation of $PM_{2.5}$ in Beijing
under the influence of transport-induced pollution events (ambient $PM_{2.5}$ exceeding both local
$PM_{2.5}$ and 75 µg m$^{-3}$) shows a general trend of high concentrations in winter and low
concentrations in summer. The main reason for this phenomenon is related to the seasonal



emissions of pollutants in China and the characteristics of climate change. Before 2019, the
west region was the primary source of pollution events during autumn and winter. However,
starting from 2019, the east region became the main contributor of polluted air masses in
autumn. Additionally, among all regions, the east region exhibited the smallest decrease in
transport-induced pollution events after 2017.
From 2013 to 2020, local emissions were the main contributors to pollution events in Beijing.
However, the Air Pollution Prevention and Control Action Plan, implemented by the Chinese
government in 2017, more effectively mitigated pollutants caused by regional transport
compared to local emissions, particularly during autumn and winter. This finding suggests that
Beijing should prioritize reducing local emissions while also accounting for potential
contributions from the east region in its future pollution prevention and control strategies.

**Code and data availability**
The Machine learning code is archived on Zenodo at https://doi.org/10.5281/zenodo.13994450,
while the CWT code is archived on Zenodo at https://doi.org/10.5281/zenodo.13994400. The
meteorology and $PM_{2.5}$ data used in this study can be accessed at
https://dx.doi.org/10.17632/bhfktx3kz8.2.

**Author contribution**
Kang Hu, Hong Liao and Dantong Liu designed and carried out the experiments. Kang Hu
wrote the code and final paper with contributions from all other authors. Hong Liao, Dantong
Liu, Lei Chen and Jianbing Jin reviewed and edited the paper.

**Competing interests**
The contact author has declared that none of the authors has any competing interests.

**Acknowledgements**
This research was supported by the China Postdoctoral Science Foundation (2023M741773),
Postdoctoral Fellowship Program of CPSF (GZC20231150).

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

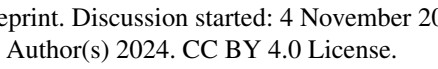

## Figures and captions

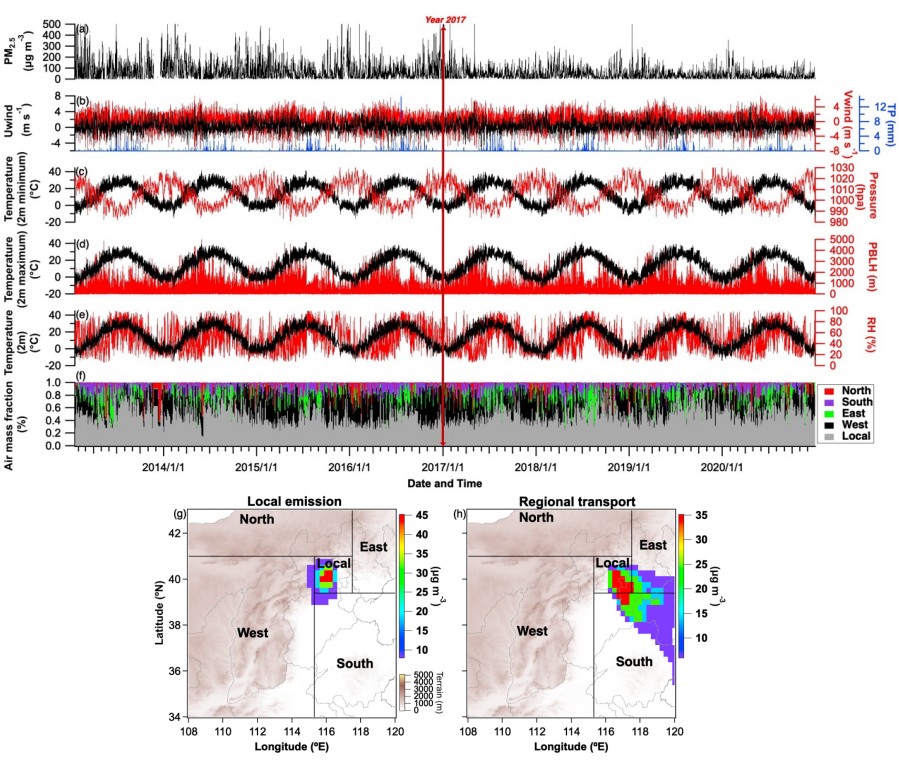

Fig. 1. Temporal evolution of parameters used in the XGBoost model: (a) PM$_{2.5}$; (b) U-wind,

V-wind, and total precipitation; (c) 2-m minimum temperature and surface pressure; (d) 2-m

maximum temperature and planetary boundary layer height; (e) 2-m temperature and relative

humidity; (f) air mass fraction in contributing sources derived from the Concentration-

Weighted Trajectory (CWT) model for a 1-day backward trajectory. The red vertical line with

arrows indicates the implementation of environmental regulations. Typical examples of the

CWT model analysis are shown for (g) a local emission period (25 August 2013) and (h) a

regional transport period (15 July 2013).



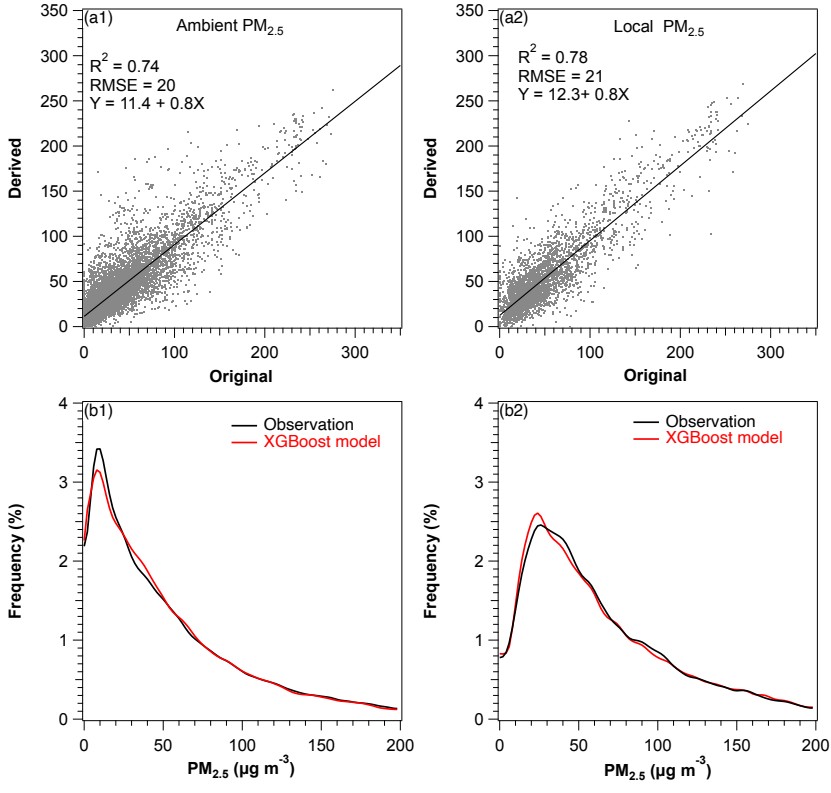

520

Fig. 2. Comparison of XGBoost model estimates and measurements for (a1) ambient $PM_{2.5}$ and

(a2) local $PM_{2.5}$ using testing samples from 2020. Frequency distributions of $PM_{2.5}$ observations

(black lines) and XGBoost model predictions (red lines) obtained through fivefold cross-

validation for (b1) ambient $PM_{2.5}$ and (b2) local $PM_{2.5}$.

525



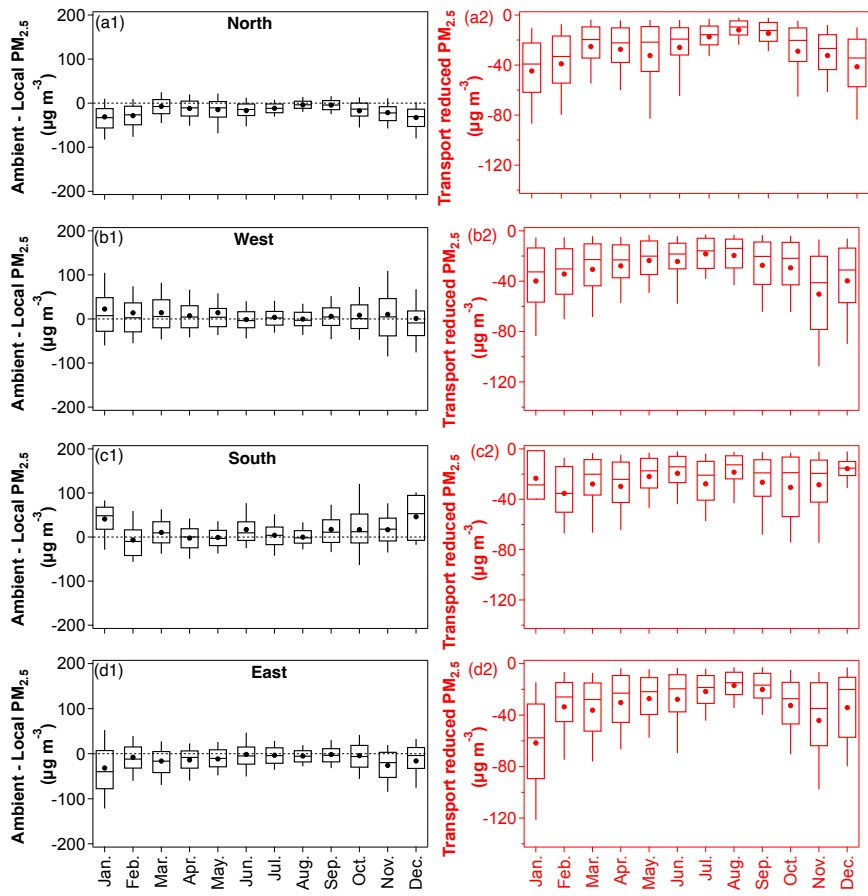

Fig. 3. Monthly variations of the difference between ambient and local $PM_{2.5}$ from the (a1) North, (b1) West, (c1) South, and (d1) East regions. Right panels show monthly variations of $PM_{2.5}$ reductions caused by regional transport for the corresponding source regions in the left panels. The upper and lower boundaries represent the 75$^{th}$ and 25$^{th}$ percentiles, respectively, while the solid origin represents the average value.



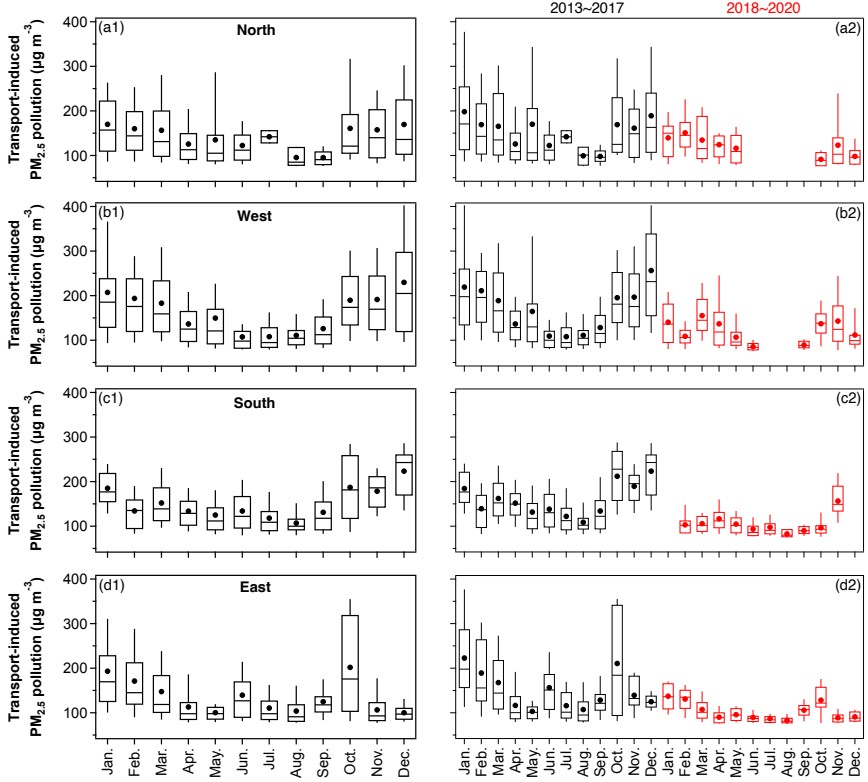

Fig. 4. Monthly variations of transport-induced $PM_{2.5}$ pollution (ambient $PM_{2.5}$ exceeding local $PM_{2.5}$ and 75 µg m$^{-3}$) from the (a1) North, (b1) West, (c1) South, and (d1) East regions during 2013-2020. Right panels show monthly variations of transport-induced $PM_{2.5}$ pollution before (black) and after (red) 2017 for the corresponding source regions in the left panels. The upper and lower boundaries represent the 75th and 25th percentiles, respectively, while the solid origin represents the average result.



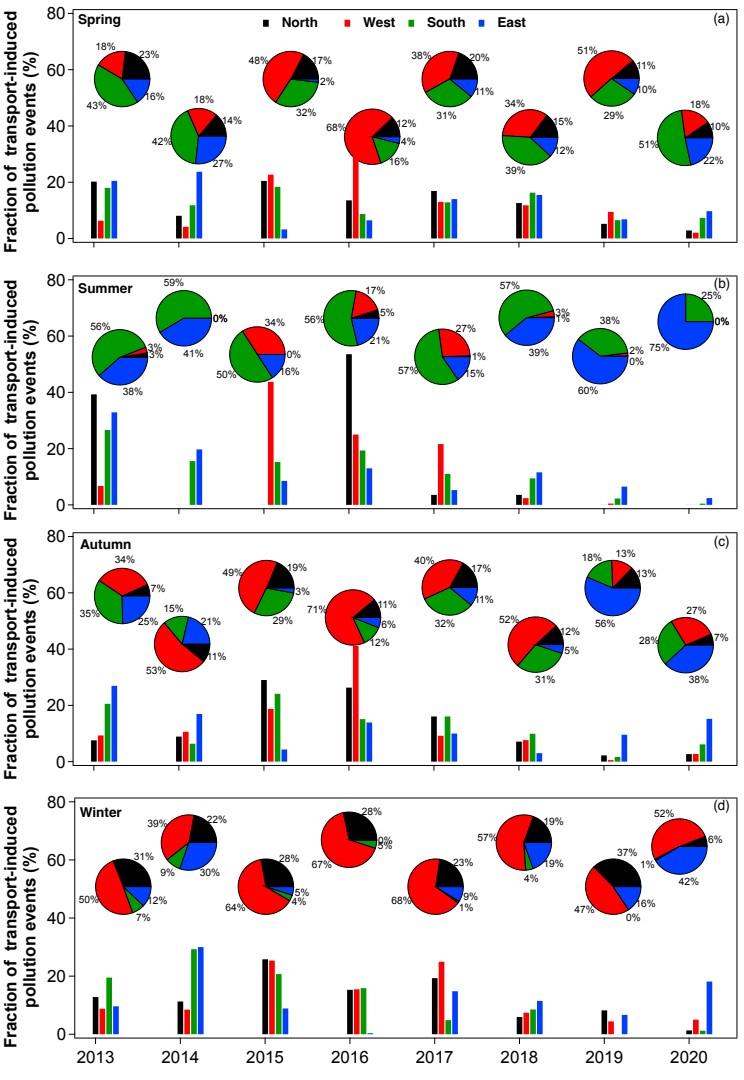

541

Fig. 5. Histograms depict the annual fraction of transport-induced pollution events in each

direction relative to the total number of occurrences from 2013 to 2020 during (a) spring, (b)

summer, (c) autumn, and (d) winter. Pie charts illustrate the proportion of transport-induced

pollution events in each direction for each year within the corresponding seasons.

546