# Peer review of "A Novel Method for Quantifying the Contribution of Regional Transport to PM2.5 in Beijing"

_Geoscientific Model Development, 2024_

## Referee Comment (RC1)

This study combined the HYSPLIT ensemble with CWT to obtain hourly resolution of pollutant sources and employed machine learning method to quantify the contributions of local emissions and regional transport in Beijing. The article highlights that local emissions were the main cause of pollution events in Beijing from 2013 to 2020 and that the Air Pollution Prevention and Control Action Plan had a more significant effect on reducing emissions through regional transmission. After addressing the following comments, I believe this work has excellent potential for publication.

General Comments

1. Line 43: "was gradually increasing"

2. Line 44: "contributors to"

3. Line 122-127: Compared to previous studies that relied solely on CWT analysis with HYSPLIT trajectories to distinguish between local emissions and regional transport, what specific improvements does your study introduce? In other words, after integrating XGBoost models, what are the advantages of your approach in enhancing the analysis? What specific problems or limitations of the previous methods does your study address? These aspects should be clearly articulated to highlight the improvements and contributions of your work.

4. Line 142: "which are available every 3 hours."

5. Lines 148 to 160: There is an issue with the formula used to calculate the potential source region airflow trajectory weight concentration using CWT. $C_{ij}$ represents the average weight concentration of the ij-th grid, and $W_{ij}$ is the weight coefficient of grid $(i,j)$ used to reduce uncertainty. Therefore, there is no need to multiply by $W_{ij}$ when calculating $C_{ij}$; multiplying by $W_{ij}$ is for calculating WCWT.

6. Line 177-180: The authors employed the XGBoost model to predict $PM_{2.5}$ concentrations, using only meteorological, temporal variables and PBLH as input parameters. Considering that the data in this study were obtained from national monitoring stations, which typically provide detailed information on conventional pollutants (e.g., $PM_{10}$, $SO_2$, $NO_x$, $O_3$, CO), would the exclusion of these pollutant data from the model input impact the model's performance?

7. Line 185-195: I'm very interested in how the authors used the XGBoost model to separate local $PM_{2.5}$ from ambient $PM_{2.5}$, as this could be incredibly valuable for work in this field. However, the explanation in this section lacks sufficient detail on how this was achieved. I

believe other readers might have similar questions. It would be both helpful and necessary if the authors could provide more detailed and clear explanations to make the paper easier to understand and more applicable.

8. Line 215: The sentence "Fig. S2 reveals that ambient pollution events ($PM_{2.5}$>75 µg m$^{-3}$) in Beijing are primarily influenced by air masses originating from the south and west, particularly under the control of westward air masses." It merely presents the observed phenomenon that ambient pollution events in Beijing are mainly affected by air masses from the south and west, especially under the influence of westward air masses, but fails to provide in-depth analysis or explanation for why the westward air masses have a stronger influence in certain circumstances. It lacks speculation or reference to relevant studies to enhance the understanding of the underlying reasons for this phenomenon.

9. Line 303: "from the south region"

10. Why does the manuscript divide the year into four seasons (spring, summer, autumn, and winter) instead of four quarters? The commonly understood seasons have time differences, and given the long-time span of this study, this could introduce some error. The study needs to clearly define how spring, summer, autumn, and winter are defined each year.

11. Based on the CWT combined with the HYSPLIT ensemble, the authors distinguished between local emissions and regional transport. However, in the subsequent machine learning process, the authors used XGBoost to derive locally emitted $PM_{2.5}$ and then derived the regionally transported $PM_{2.5}$. Why choose this approach instead of learning regional transmission to calculate local emissions? Please explain the reasoning.

---

## Author Comment (AC1)

Dear Editor,

We thank reviewers for their constructive comments which have greatly improved our manuscript. We have now addressed all comments reviewers raised.

Referee: 1

This study combined the HYSPLIT ensemble with CWT to obtain hourly resolution of pollutant sources and employed machine learning method to quantify the contributions of local emissions and regional transport in Beijing. The article highlights that local emissions were the main cause of pollution events in Beijing from 2013 to 2020 and that the Air Pollution Prevention and Control Action Plan had a more significant effect on reducing emissions through regional transmission. After addressing the following comments, I believe this work has excellent potential for publication.

We are thankful for the valuable comments on our work from the reviewer.

General Comments

1.  Line 43: "was gradually increasing"

This is now has been revised.

2.  Line 44: "contributors to"

This is now has been revised.

3.  Line 122-127: Compared to previous studies that relied solely on CWT analysis with HYSPLIT trajectories to distinguish between local emissions and regional transport, what specific improvements does your study introduce? In other words, after integrating XGBoost models, what are the advantages of your approach in enhancing the analysis? What specific problems or limitations of the previous methods does your study address? These aspects should be clearly articulated to highlight the improvements and contributions of your work.

We thank reviewer to point this out. This is now added in the revision:

Line 125-127: "In this study, we combined CWT analysis with the HYSPLIT trajectory ensemble to obtain hourly-resolution $PM_{2.5}$ source results and used this approach to distinguish between local emissions and regional transport. Solved the problems of traditional CWT methods being unable to obtain hourly time accuracy and models such as NAME consuming a large number of computational resources."

4. Line 125: The sentence "Fig. S2 reveals that ambient pollution events ($PM_{2.5}>75$ µg m$^{-3}$) in Beijing are primarily influenced by air masses originating from the south and west, particularly under the control of westward air masses." It merely presents the observed phenomenon that ambient pollution events in Beijing are mainly affected by air masses from the south and west, especially under the influence of westward air masses, but fails to provide in-depth analysis or explanation for why the westward air masses have a stronger influence in certain circumstances. It lacks speculation or reference to relevant studies to enhance the understanding of the underlying reasons for this phenomenon.

We thank reviewer to point this out. This is now added in the revision:

Line 229-241: "Numerous studies have indicated that air masses originating from the western region significantly contribute to regional pollution events in Beijing (Streets et al., 2007; Tian et al., 2019; Liu et al., 2020)"

5. Line 142: "which are available every 3 hours."

This is now has been revised.

6. Lines 148 to 160: There is an issue with the formula used to calculate the potential source region airflow trajectory weight concentration using CWT. $C_{ij}$ represents the average weight concentration of the ij-th grid, and $W_{ij}$ is the weight coefficient of grid (i,j) used to reduce uncertainty. Therefore, there is no need to multiply by $W_{ij}$ when calculating $C_{ij}$; multiplying by $W_{ij}$ is for calculating WCWT.

We thank reviewer to point this out. This is now added in the revision:

Line 156-171: "In the CWT analysis method, each grid point is assigned a weight, and the contribution of each grid point to the pollutant concentration at the target site is calculated using the air mass residence time and pollutant concentration (Hopke et al., 1993; Polissar et al., 1999; Xu and Akhtar, 2010) (equation 1). The grid point resolution was set to 0.25°×0.25° for this study. In equations 1, $C_{ij}$ is the average weighted concentration at grid point (i, j), l is the trajectory index, M represents the total number of trajectories, $C_l$ is the $PM_{2.5}$ concentration corresponding to the target site, and $\tau_{ijl}$ is the residence time of trajectory l passing through the grid point. In calculation, the number of trajectories falling on each grid point is used instead of the residence time.

$$C_{ij} = \frac{\sum_{l=1}^{M} C_l \times \tau_{ijl}}{\sum_{l=1}^{M} \tau_{ijl}} \tag{1}$$

To reduce the effect of small values of $n_{ij}$, the CWT values were multiplied by an arbitrary weight function $W(n_{i,j})$ to better reflect the uncertainty in the values for these grids (equation 2).

$$W(n_{i,j}) = \begin{cases} 1.00, & 3n_{ave} < n_{ij} \\ 0.70, & 1.5n_{ave} < n_{ij} \leq 3n_{ave} \\ 0.4, & n_{ave} < n_{ij} \leq 1.5n_{ave} \\ 0.17, & n_{ij} \leq n_{ave} \end{cases} \tag{2}$$

where $n_{ij}$ represents the number of trajectories that fall within the grid point, and $n_{ave}$ represents the average number of trajectories passing through each grid point."

7. Line 177-180: The authors employed the XGBoost model to predict PM$_{2.5}$ concentrations, using only meteorological, temporal variables and PBLH as input parameters. Considering that the data in this study were obtained from national monitoring stations, which typically provide detailed information on conventional pollutants (e.g., PM$_{10}$, SO$_2$, NO$_x$, O$_3$, CO), would the exclusion of these pollutant data from the model input impact the model's performance?

This study chose to use only meteorological data to learn PM$_{2.5}$ for two main reasons. Firstly, the learned PM$_{2.5}$ values include both the ambient and the locally emitted PM$_{2.5}$ values. Incorporating ambient PM or AOD values into the machine learning process may impact the local emission results. Secondly, numerous studies have confirmed that PM values can be obtained using meteorological data combined with machine learning method.

It is worth noting that many studies also use meteorological data combined with PM or AOD values to learn and obtain actual atmospheric PM results. For example, Xiao et al. used AOD combined with meteorological data to learn PM$_{2.5}$, achieving an $r^2$ result around 0.8 (Xiao et al., 2021). Similarly, Xu et al. used model-provided PM$_{2.5}$ combined with meteorological data to learn ambient PM$_{2.5}$ values, obtaining an $r^2$ result around 0.91 (Xu et al., 2023). However, despite the addition of AOD and PM parameters, there is still a significant difference in the $r^2$ values obtained from these studies, suggesting that sufficient training data is another important factor affecting the learning results.

In this study, the ambient and local PM$_{2.5}$ emissions obtained from meteorological data were compared with actual observations, yielding $r^2$ values of 0.74 and 0.78, respectively. These learning results are considered acceptable for the purposes of this study.

8. Line 185-195: I'm very interested in how the authors used the XGBoost model to separate local PM$_{2.5}$ from ambient PM$_{2.5}$, as this could be incredibly valuable for work in this field. However, the explanation in this section lacks sufficient detail on how this was achieved. I

believe other readers might have similar questions. It would be both helpful and necessary if the authors could provide more detailed and clear explanations to make the paper easier to understand and more applicable.

We thank reviewer to point this out. This is now added in the revision:

Line 128-133: "By training the XGBoost model with $PM_{2.5}$ dominated by local emissions, which are separately distinguished by CWT, and generalizing the findings to all study periods, the concentration of locally emitted $PM_{2.5}$ (local) can be obtained. Similarly, ambient observed $PM_{2.5}$ (ambient) can be determined by training the XGBoost model with ambient $PM_{2.5}$ data. The contribution of regional transport to $PM_{2.5}$ in Beijing can be quantified by comparing the ambient and local $PM_{2.5}$ concentrations."

9. Line 303: "from the south region"

This is now has been revised.

10. Why does the manuscript divide the year into four seasons (spring, summer, autumn, and winter) instead of four quarters? The commonly understood seasons have time differences, and given the long-time span of this study, this could introduce some error. The study needs to clearly define how spring, summer, autumn, and winter are defined each year.

We thank reviewer to point this out. This is now added in the revision:

Line 143-145: "In this study, a year was divided into four quarters: Spring (March, April, and May), Summer (June, July, and August), Autumn (September, October, and November), and Winter (December, January, and February)."

11. Based on the CWT combined with the HYSPLIT ensemble, the authors distinguished between local emissions and regional transport. However, in the subsequent machine learning process, the authors used XGBoost to derive locally emitted $PM_{2.5}$ and then derived the regionally transported $PM_{2.5}$. Why choose this approach instead of learning regional transmission to calculate local emissions? Please explain the reasoning.

Local emission sources in Beijing have more stable pollution components compared to regional transmission. Thus, the results obtained from learning local emission sources are believed to be more consistent with actual observed values compared to regional emissions, which are influenced by various sources. Therefore, in this study, regional transport contributions are determined by subtracting local emissions from the ambient concentrations, rather than learning regional transport and calculating local emission values.

**References**

Hopke, P. K., Gao, N., and Cheng, M.-D.: Combining chemical and meteorological data to infer source areas of airborne pollutants, Chemometrics and Intelligent Laboratory Systems, 19, 187-199, 1993.

Liu, D., Hu, K., Zhao, D., Ding, S., Wu, Y., Zhou, C., Yu, C., Tian, P., Liu, Q., and Bi, K.: Efficient vertical transport of black carbon in the planetary boundary layer, Geophysical Research Letters, 47, e2020GL088858, 2020.

Polissar, A., Hopke, P., Paatero, P., Kaufmann, Y., Hall, D., Bodhaine, B., Dutton, E., and Harris, J.: The aerosol at Barrow, Alaska: long-term trends and source locations, Atmospheric Environment, 33, 2441-2458, 1999.

Streets, D. G., Fu, J. S., Jang, C. J., Hao, J., He, K., Tang, X., Zhang, Y., Wang, Z., Li, Z., and Zhang, Q.: Air quality during the 2008 Beijing Olympic games, Atmospheric environment, 41, 480-492, 2007.

Tian, P., Liu, D., Huang, M., Liu, Q., Zhao, D., Ran, L., Deng, Z., Wu, Y., Fu, S., and Bi, K.: The evolution of an aerosol event observed from aircraft in Beijing: An insight into regional pollution transport, Atmospheric Environment, 206, 11-20, 2019.

Xiao, Q., Zheng, Y., Geng, G., Chen, C., Huang, X., Che, H., Zhang, X., He, K., and Zhang, Q.: Separating emission and meteorological contributions to long-term PM2.5 trends over eastern China during 2000–2018, Atmospheric Chemistry and Physics, 21, 9475-9496, 10.5194/acp-21-9475-2021, 2021.

Xu, R., Ye, T., Yue, X., Yang, Z., Yu, W., Zhang, Y., Bell, M. L., Morawska, L., Yu, P., and Zhang, Y.: Global population exposure to landscape fire air pollution from 2000 to 2019, Nature, 621, 521-529, 2023.

Xu, X. and Akhtar, U.: Identification of potential regional sources of atmospheric total gaseous mercury in Windsor, Ontario, Canada using hybrid receptor modeling, Atmospheric Chemistry and Physics, 10, 7073-7083, 2010.

---

## Author Comment (AC2)

Dear Editor,

We thank reviewers for their constructive comments which have greatly improved our manuscript. We have now addressed all comments reviewers raised.

Referee: 2

This is a very interesting study estimating the contribution of regional transport to PM2.5 in Beijing. This analysis can support policy-makers in both validating and designing effective policies. In addition, the methodology can be potentially applied in other regions as well, unraveling the contribution of local emissions and regional pollution transport. Nevertheless, there are some points that need further information and clarifications. If these points are addressed, then I would be happy to suggest publication in GMD.

We are thankful for the generally positive comments on our work from the reviewer.

**Main comments**

I.    A better description of the XGBoost model goal is needed. Is there a separate XGBoost model for ambient and local? Which exactly is the use of term ambient in this study? Please explain in more detail what the difference (in train data) is in building these two models. Which are the features and which are the targets in each case? One should clearly understand the transition from HYSPLIT CWT analysis to the XGBoost models. Which information from the HYSPLIT and CWT analysis are used and in which way to build the XGBoost models, and for which specific reasons (goals).

We thank reviewer to point this out. These are now added in the revision:

Line 123-133: "In this study, we combined CWT analysis with the HYSPLIT trajectory ensemble to obtain hourly-resolution $PM_{2.5}$ source results and used this approach to distinguish between local emissions and regional transport. Solved the problems of traditional CWT methods being unable to obtain hourly time accuracy and models such as NAME consuming a large number of computational resources. Predictive XGBoost models were developed for Beijing using meteorological data and time variables to explain $PM_{2.5}$ concentrations. By training the XGBoost model with $PM_{2.5}$ dominated by local emissions, which are separately distinguished by CWT, and generalizing the findings to all study periods, the concentration of locally emitted $PM_{2.5}$ (local) can be obtained. Similarly, ambient observed $PM_{2.5}$ (ambient) can be determined by training the XGBoost model with ambient $PM_{2.5}$ data. The contribution of regional transport to $PM_{2.5}$ in Beijing can be quantified by comparing the ambient and local $PM_{2.5}$ concentrations."

II.    The year 2020 is used in the analysis and to validate the XGBoost model training. Yet, 2020 is a "special" year due to COVID-19 and the associated restriction measures having direct effect on emissions and air pollution levels. How do COVID-19 restriction measures affect the analysis, and the conclusions raised for the Action Plan? This is something that needs to be well clarified.

We thank reviewer to point this out. In this study, the $r^2$ for ambient $PM_{2.5}$ and local emissions reached 0.78 and 0.74, respectively. The relatively low $r^2$ values may be attributed to the extremely low human activities under the same meteorological conditions, which led to a decrease in $PM_{2.5}$ concentration, making it challenging for XGBoost to learn better results. As shown in Figure 2b1 and b2, the significant differences between XGBoost learning results and actual observations are mainly concentrated in the low $PM_{2.5}$ concentration stage. To address this, we have added the following description in the article:

Line 228-232: "The error between XGBoost learning results and actual observed $PM_{2.5}$ values is mainly concentrated in the low concentration stage. This may be attributed to the significant reduction in human activities during the COVID-19 lockdown periods, which led to a decrease in actual $PM_{2.5}$ levels, making it challenging for XGBoost to learn (Fig. 2b1 and b2)."

Line 571-575: "

[Figure]

Fig. 2. Comparison of XGBoost model estimates and observations for (a1) ambient $PM_{2.5}$ and (a2) local $PM_{2.5}$ using testing samples from 2020. Frequency distributions of $PM_{2.5}$ observations

(black lines) and XGBoost model predictions (red lines) for (b1) ambient PM$_{2.5}$ and (b2) local PM$_{2.5}$ using testing samples from 2020."

Furthermore, the inconsistent lockdown times of various cities during the COVID-19 pandemic have impacted the calculation of ambient and local PM$_{2.5}$ during 2020. To clarify this, we have added the following explanation:

Line 199-200: "Note that the 2020 analysis results may contain some uncertainties due to the impact of COVID-19."

Line 312-315: "The highest proportion of regional transport events from the west occurred in 2016, reaching 68%, while the highest proportion of southward transport-induced pollution events occurred in 2017 (with the exception of 2020, which may have been influenced by the COVID-19 pandemic)."

Line 325-328: "However, after the implementation of the Action Plan, the proportion of transport-induced pollution events from the south region gradually decreased to 38%. In 2020, this proportion further declined to 25%, but this may have been affected by the COVID-19 pandemic."

III. Do you apply any "feature selection" based on feature importance? What is the rationale for using both month of the year and day of the year? How is overfitting prevented in the XGBoost model training? Several studies (e.g. Akritidis et al., 2021; Zhang et al., 2020) applied an early stopping technique to prevent overfitting. Is there something similar applied here? Please explain and discuss in the manuscript accordingly.

The selection of feature importance parameters in our article primarily referred to the work by Xu et al., published in Nature, who have established mature learnable feature importance parameters for PM$_{2.5}$ (Xu et al., 2023). The main goal of our study is to obtain machine learning PM$_{2.5}$ results that match the observed values; therefore, less attention was paid to the choice of feature importance parameters. Additionally, we have added the contribution of feature importance in the revised manuscript, as follows:

Line 196-197: "Based on the XGBoost learning results, the most sensitive parameters for both local and ambient PM$_{2.5}$ are RH, wind field, surface pressure and PBLH (Fig. S1)."

Line 30-35 in supplement:

[Figure]

Fig. S1. Feature importance of the XGBoost models in estimating local (black) and ambient (green) $PM_{2.5}$ using the training data. The considered features include temperature, 2-m maximum (mx2t), 2-m minimum temperature (mn2t), relative humidity (RH), surface pressure, V-wind, U-wind, planetary boundary layer height (PBLH), total precipitation (tp), year, day of year, day of week, and month."

The 'month' feature can explain monthly variation characteristics. However, there is a jump in month values between 1 and 12, which may lead to inconsistencies when directly using 'month' as a numerical feature, affecting the model's learning of periodic patterns. The 'day of year' feature can capture the characteristics of certain small-scale days, such as those before and after the Spring Festival or during a specific summer vacation period. When using both 'month' and 'day of year' features to solve scenarios with significant seasonality (monthly/quarterly changes) and differences within a month, the model can capture cross-month trends and subtle differences in specific dates within the month.

Thanks for pointing out the missing parts in our article. We have added the overfitting prevention methods used during the training process of the XGBoost model, as follows:

Line 212-214: "Hyperparameter optimization and performance evaluation of the model were conducted using fivefold cross-validation (CV), while early stopping with a patience of 10 rounds was employed to prevent overfitting (Akritidis et al., 2021; Zhang et al., 2020)."

**Comments**

1. Line 53: I believe dust deserves to be included compared to tsunamis and volcanic eruptions.

We thank reviewer to point this out. This is now added in the revision:

Line 52-55: "Ambient fine particulate matter (PM$_{2.5}$, with particle aerodynamic diameter ≤ 2.5 μm) is influenced by both natural sources, such as dust, volcanic eruptions, tsunamis, and forest fires, and anthropogenic emissions, including fuel combustion, transportation, and industrial production."

2. Line 59: The more recent studies by Smith et al. (2020) and Kalisoras et al. (2024) based on CMIP6 ESMs can be also included here (see details in References).

This is now added in the revision:

Line 56-59: "Numerous epidemiological studies have found that PM$_{2.5}$ can significantly damage human health by exacerbating respiratory and cardiovascular diseases (Bartell et al., 2013; Brauer et al., 2012; Pascal et al., 2014), and also has an impact on weather and climate change (Wang et al., 2014; Smith et al., 2020; Kalisoras et al., 2023)."

3. Lines 60-61: The study by Geng et al. (2021) can be also cited here.

This is now added in the revision:

Line 59-61: "China's rapid and energy-intensive development over the past several decades has led to severe air pollution and negative public health impacts (Huang et al., 2014; Geng et al., 2021)."

4. Lines 68-69: For this statement a reference is required.

We thank reviewer to point this out. This is now added in the revision:

Line 66-70: "To address this issue, the Chinese government implemented the Air Pollution Prevention and Control Action Plan (denoted "Action Plan") from 2013 to 2017 and the Blue Sky Protection Campaign from 2018 to 2020, which effectively controlled anthropogenic emissions and reduced ambient PM$_{2.5}$ concentrations (Zhang et al., 2019; Du et al., 2022)."

5. Line 72: Change to Wu et al. (2021) and apply accordingly where applicable in the manuscript.

We thank reviewer to point this out. This is now added in the revision:

Line 73-76: "Wu et al. used the HYSPLIT model to simulate the 24-hour backward trajectory in Zhoushan (Wu et al., 2021), and identified continental air masses that spent more than 5% of the previous 24 hours over the continent region, while the remaining air masses were identified as oceanic-influenced air masses."

Furthermore, we have corrected the corresponding errors throughout the manuscript.

6. Lines 116-117: "meteorological data, synoptic scale, planetary boundary layer height (PBLH)," What do you mean by synoptic scale? PBLH can be considered a meteorological parameter as well. Please rephrase.

We thank reviewer to point this out. This is now added in the revision:

Line 117-119: "For instance, Grange et al. used meteorological data, synoptic scale weather patterns, and time variables to explain daily $PM_{10}$ concentrations in Switzerland (Grange et al., 2018)."

7. I suggest listing the selected hyperparameters for each XGBoost model to facilitate reproduction if needed.

We thank reviewer to point this out. This is now added in the revision:

Line 185-189: "The hyperparameters used in the model for local (ambient) conditions include a maximum number of boosting iterations of 6067 (13421), a learning rate of 0.1, a maximum tree depth of 7 (11), a minimum sum of instance weight needed in a child of 5 (3), a subsampling ratio of 0.8 (0.6) for training instances, and a subsampling ratio of 0.8 for columns when constructing each tree."

8. Lines 131-132: A url and/or reference for the $PM_{2.5}$ observations is needed here.

We have added the $PM_{2.5}$ data to the Code and Data Availability section.

Line368-375: "The codes used in this study are archived on Zenodo: the machine learning code at https://doi.org/10.5281/zenodo.14677125, the CWT code at https://doi.org/10.5281/zenodo.13994400, ECMWF data at https://doi.org/10.5281/zenodo.14353871, GDAS data at https://doi.org/10.5281/zenodo.14347277, HySplit Trajectory Ensemble at https://doi.org/10.5281/zenodo.14375567, and PySPLIT at https://doi.org/10.5281/zenodo.14354765. The meteorology and $PM_{2.5}$ data used in this study can be accessed at https://dx.doi.org/10.17632/bhfktx3kz8.2."

9. Line 136: A reference is needed here for ERA5 data set.

We thank reviewer to point this out. This is now added in the revision:

Line 140-143: "Meteorological data, including temperature, relative humidity, pressure, precipitation, wind speed, and planetary boundary layer height (PBLH), were sourced from the European Centre for Medium-Range Weather Forecasts (ECMWF) ERA5 hourly reanalysis dataset (https://cds.climate.copernicus.eu/datasets)."

10. Lines 148-160: Equation 2 is referred first. Please either refer to equation 1 first or change the order of equations.

We thank reviewer to point this out. We have made the following modifications:

Line 156-171: "In the CWT analysis method, each grid point is assigned a weight, and the contribution of each grid point to the pollutant concentration at the target site is calculated using the air mass residence time and pollutant concentration (Hopke et al., 1993; Polissar et al., 1999; Xu and Akhtar, 2010) (equation 1). The grid point resolution was set to 0.25°×0.25° for this study. In equations 1, $C_{ij}$ is the average weighted concentration at grid point $(i, j)$, $l$ is the trajectory index, $M$ represents the total number of trajectories, $C_l$ is the PM$_{2.5}$ concentration corresponding to the target site, and $\tau_{ijl}$ is the residence time of trajectory $l$ passing through the grid point. In calculation, the number of trajectories falling on each grid point is used instead of the residence time.

$$C_{ij} = \frac{\sum_{l=1}^{M} C_l \times \tau_{ijl}}{\sum_{l=1}^{M} \tau_{ijl}} \tag{1}$$

To reduce the effect of small values of $n_{ij}$, the CWT values were multiplied by an arbitrary weight function $W(n_{i,j})$ to better reflect the uncertainty in the values for these grids (equation 2).

$$W(n_{i,j}) = \begin{cases} 1.00, & 3n_{ave} < n_{ij} \\ 0.70, & 1.5n_{ave} < n_{ij} \le 3n_{ave} \\ 0.4, & n_{ave} < n_{ij} \le 1.5n_{ave} \\ 0.17, & n_{ij} \le n_{ave} \end{cases} \tag{2}$$

where $n_{ij}$ represents the number of trajectories that fall within the grid point, and $n_{ave}$ represents the average number of trajectories passing through each grid point."

11. Lines 154-156: I am a bit confused here. First you say "$\tau$ijl is the residence time of trajectory l passing through the grid point" and then "In calculation, the number of trajectories falling on each grid point is used instead of the residence time". The residence time is calculated for each trajectory, but then how can a residence time for a trajectory be calculated from a number of trajectories? I may miss something here, please clarify.

The calculation process only uses the number of times a trajectory passes through a specific grid point instead of counting the trajectory's residence time within that grid point.

12. Lines 162-166: Which is the rationale behind the definition of the regions? I think a small sentence is needed.

We thank reviewer to point this out. This is now added in the revision:

Line 172-173: "The potential source contribution to $PM_{2.5}$ at the target site was investigated by categorizing the backward air masses into five different source regions centered around Beijing:"

13. Lines 168-170: So, if the contribution from the north sector is 41% and the one from local is 40% then is classified as regional from the north sector?

We thank reviewer to point this out. This study considers the dominant air mass to be the main incoming air mass. This method has been applied in previous studies (Streets et al., 2007; Tian et al., 2019; Liu et al., 2020). However, these studies only consider westward air masses as regional transport processes. To explore the pollution and cleaning effects of regional transport, we have improved the classification method by using the dominant air mass. We have made the following modifications to the corresponding parts of the article:

Line 179-182: "The region with the highest contribution is used to determine the dominant source of air masses in Beijing at each time, classifying the overall air mass sources into local emissions (Fig. 1g) and regional transport (Fig. 1h). It is important to note that local emission periods were also influenced by persistent regional transport, and vice versa."

14. Figure 5: Just for clarification, summing the individual histograms over the years will result in100%? In some cases, the pie charts sum is not 100%. I assume this is related to the rounding of percentages.

We have verified the summation results. For sums that deviate from 100%, the error values are consistently 1% due to rounding.

[revised manuscript text omitted]

---

## Author Response (AR2)

Dear Editor,

We have now addressed all comments reviewers raised.

Referee:

I appreciate the effort made by the authors to address the comments, which they mostly did. I have only two comments regarding the revised version.

We are thankful for the valuable comments from the reviewer.

1. Please clarify what has changed in the revised Figure 2 compared to the previous version. What do the authors mean with the term "testing samples"? What is the size of it? Why don't they include the whole year (or they do)? COVID-19 restriction measures began (roughly) at late January and early February. I suggest including (like Figure 2) two testing samples as different colors. As PRE COVID-19 period all samples from January 2020 and as COVID-19 period February to December 2020 samples. This might depict the effect of COVID-19 in XGBoost model performance. I understand that this might be out of the scope of the study, but this can be included as Supplement to better assess the COVID-19 effect.

Figure 2 originally used the data from the entire period to do frequency analysis. However, the current version only includes the results from the testing data.

We thank the reviewer for their comments on our wording. We have modified the corresponding content in the manuscript. Additionally, to avoid misunderstanding regarding the testing data period, we have updated the caption in Figure 2 by removing "2020" and using the unified "testing data".

Line 197-200: "For the machine learning process, data from 2013 to 2019 were used for training the XGBoost models, while the 8613 data points measured from January 1 to December 31, 2020, were used for model testing (Fig. S2). Note that the 2020 analysis results may contain some uncertainties due to the impact of COVID-19."

[Figure]

Fig. 2. Comparison of XGBoost model estimates and observations for (a1) ambient $PM_{2.5}$ and (a2) local $PM_{2.5}$ using testing data. Frequency distributions of $PM_{2.5}$ observations (black lines) and XGBoost model predictions (red lines) for (b1) ambient $PM_{2.5}$ and (b2) local $PM_{2.5}$ using testing data.

We have added the following figure to the revised manuscript's supplementary material:

[Figure]

Fig. S2. Temporal evolution of $PM_{2.5}$ observations (black lines) and XGBoost model predictions (red lines) using testing data. The grey and blue shades represent the periods pre-COVID-19 and during COVID-19, respectively.

2.  In accordance with the new discussion included "This may be attributed to the significant reduction in human activities during the COVID-19 lock-down periods, which led to a decrease in actual PM2.5 levels." a phrase for the fact that COVID-19 restriction measures might have contributed to the reduced PM2.5 levels of 2020 should be included in the Abstract and the Conclusions as well.

3.  We thank reviewer to point this out. These are now added in the revision:

Line 43-46: "The COVID-19 restrictions might have reduced $PM_{2.5}$ concentrations in 2020. From 2013 to 2020, local emissions were the main contributors to pollution events in Beijing. The Action Plan has more effectively reduced pollution caused by regional transport, particularly during autumn and winter."

Line 349-353: "These clean air masses from different directions exhibited similar seasonal variations in their ability to reduce ambient pollution in Beijing, with a stronger reduction effect in winter and a weaker reduction effect in summer. In addition to clean air masses, COVID-19 restrictions might have contributed to the reduction of $PM_{2.5}$ in 2020."